# Habitat use of loggerhead (*Caretta caretta*) and green (*Chelonia mydas*) turtles at the northern limit of their distribution range of the Northwest Pacific Ocean

Il-Hun Kim[1]☯, Il-Kook Park[2]☯, Daesik Park[2], Min-Seop Kim[1], In-Young Cho[1], Dongwoo Yang[1], Dong-Jin Han[3], Eunvit Cho[3], Won Joon Shim[4], Sang Hee Hong[4,5], Yong-Rock An[1]*

1 Department of Ecology and Conservation, National Marine Biodiversity Institute of Korea, Seocheon, Chungcheongnam-do, Republic of Korea, 2 Division of Science Education, Kangwon National University, Chuncheon, Gangwon-do, Republic of Korea, 3 Aqua Team, Aqua Planet Yeosu, Yeosu, Jeollanam-do, Republic of Korea, 4 Ecological Risk Research Department, Korea Institute of Ocean Science and Technology, Geoje, Gyeongsangnam-do, Republic of Korea, 5 Department of Ocean Science, University of Science and Technology, Daejeon, Republic of Korea

☯ These authors contributed equally to this work.
* rock@mabik.re.kr

**Data Availability Statement:** We have uploaded the sea turtle migration data to movebank as requested. We will compress the relevant files into

## Abstract

Verifying habitats, including the foraging and nesting areas for sea turtles, enables an understanding of their spatial ecology and successful planning of their conservation and management strategies. Recently, the observation frequency and bycatch of loggerhead (*Caretta caretta*) and green (*Chelonia mydas*) turtles have increased in the northern limit of their distribution range, in the northern part of the East China Sea and East (Japan) Sea. We conducted satellite tracking to investigate the habitat use of seven loggerhead and eight green turtles from June 2016 to August 2022 in this area, where little is known about their spatial ecology. We applied a 50 percent volume contour method to determine their main foraging areas and analyzed 6 environmental variables to characterize their habitats. Loggerhead turtles mainly stayed in and used the East China Sea as a foraging area during the tracking period, while two individuals among them also used the East Sea as a seasonal foraging area. Most green turtles also used the East China Sea as a foraging area, near South Korea and Japan, with one individual among them using the lower area of the East Sea as a seasonal foraging area. Notably, one green turtle traveled to Hainan Island in the South China Sea, a historical nesting area. Our results showed that the two sea turtle species included the East Sea as a seasonal foraging area, possibly owing to the abundance of food sources available, despite its relatively lower sea temperature. Considering that loggerhead and green sea turtles were observed using the northern part of the East China Sea and East Sea more frequently than previously known and that the sea temperature gradually increases due to climate change, conservation and management activities are required for sea turtles in these areas.

a ZIP file and submit it to the Supporting Information - Compressed/ZIP File Archive section. https://www.movebank.org/movebank/#page=studies,path=study3030282201.

**Funding:** Y-R.A. received Grant from National Marine Biodiversity Institute of Korea (2024M00300), M-S.K. received Grant from Ministry of Oceans and Fisheries (2024E00300) and W.J.S. received Grant "Development of technology for impact assessment of marine plastic debris on marine ecosystem(PE99914)" of Korea Institute of Ocean Science and Technology. The funders had no role in study design, data collection and analysis, decision to publish, or preparation of the manuscript.

**Competing interests:** The authors have declared that no competing interests exist.

## Introduction

Although many regional sea turtle populations are increasing thanks to conservation efforts [1, 2], still six out of seven known sea turtle species are threatened worldwide, which has promoted conservation and restoration efforts from many countries and global organizations, such as the Convention on International Trade in Endangered Species of Wild Fauna and Flora (CITES) and the International Union for Conservation of Nature (IUCN), are trying to conserve and restore them. Understanding the spatial and foraging ecology of sea turtles is essential for establishing successful conservation and management strategies [2–5]. However, because sea turtles have a wide distribution range, diverse habitat types, and a complex life cycle, it is necessary to invest considerable time and effort to thoroughly understand their ecology [6–8]. Currently, considering that the distribution range of marine creatures, including sea turtles, is expanding poleward owing to climate change [9–12], research on the northern edge of the sea turtle range is urgently needed. but currently insufficient. Satellite telemetry is a useful method for investigating the spatial ecology and habitat use of sea turtles, applied in various regions and species [6, 13–16]. Although various molecular ecology techniques have been recently used to study population genetic diversity and determine the studying origin of sea turtles [17–21], satellite telemetry remains a powerful study method, particularly for the distribution boundary of sea turtles, which have very low population densities [5, 14, 16, 22]. Because sea turtles show seasonal differences in habitat use, such as for foraging and nesting, and high habitat fidelity [23–26], satellite telemetry can effectively identify the major areas of these habitats [14, 27, 28]. Furthermore, analyzing these environmental characteristics of the identified habitats could reveal the factors affecting the habitat preferences of sea turtles, leading to proper conservation and management strategies against climate change and anthropogenic developments [29, 30]. Overall, the habitat information and environmental characteristics identified by satellite telemetry could contribute to the effective protection and restoration of sea turtle populations [31].

In the Northwest Pacific Ocean, loggerhead (*Caretta caretta*) and green (*Chelonia mydas*) turtles are the most commonly observed sea turtles and use the region for [32–34]. Sea turtle research in the Northwest Pacific Ocean has focused on the southern coastline of Japan and the southern Ryukyu Islands, where the main foraging and nesting sites of loggerhead and green turtles are located [19, 35–37]. Recently, the nesting frequencies of loggerhead and green turtles are declining as a consequence of coastal development and other unknown factors in these areas [18, 24, 25]. Nevertheless, the observation frequency of loggerhead and green turtles has increased in their northern distribution boundary region, including the northern parts of the East China Sea and East (Japan) Sea, near the coastline of the Korean Peninsula [34, 38, 39]. Despite the increased observation of loggerhead and green turtles at these distribution margins, studies on their habitat use are lacking. Several studies have shown that juvenile loggerhead turtles use the East Sea near South Korea and Japan as a route to the open Pacific Ocean, but few studies have been conducted on adult specimens [5, 16, 40]. In green turtles, some research showed that adults use the sea near Jeju Island [41], whereas juveniles use the Yellow Sea west of the Korean Peninsula [16]. However, the detailed patterns of habitat use by adults have not been studied yet.

In the present study, we aimed to reveal the travel patterns, habitat use, and environmental characteristics of the habitats of subadult and adult loggerhead and green turtles using satellite telemetry. Considering that the habitat use patterns of the two sea turtle species at the northern boundary of their distribution range are not well known, this study is expected to provide essential information about their spatial biology, which can be useful for the effective management and conservation of both sea turtle species in the Northwest Pacific Ocean.

## Materials and methods

### Satellite telemetry

From October 2015 to October 2022, we tracked 15 rehabilitated sea turtles (7 loggerhead and 8 green turtles), which were caught incidentally and rescued from fishing nets near the coastal area of the Republic of Korea (Table 1; S1 Table). Permissions for the sea turtle conservation program were given by the Marine Animal Protection Committee of Ministry of Oceans and Fisheries with whom this work was conducted. We measured the curved carapace length (CCL) and body weight (BW) of each sea turtle. Considering the minimum CCL of sexually mature sea turtles is known as 73 cm in loggerhead turtles and 67 cm in green turtles [42, 43], 12 turtles were identified to be adults, and three turtles as potential subadults (KOR-1, KOR-2, and KOR0155). Other than KOR0149, a male loggerhead turtle, the remaining 14 turtles were all female. The turtles were released between June and October along coast of Jeju Island (N 33.2449˚, E 126.4128˚), Yeosu (N 34.6306˚, E 127.7935˚), and Busan (N 35.1587˚, E 129.1608˚) in the Republic of Korea, where the threat of collision with ships was relatively low and the water temperature was relatively warm (16–28˚C in SST).

To track sea turtles, we used Argos satellite transmitters (Wildlife Computers Inc., Redmond, WA, USA), SPOT-352A (before September 2019, lifespan: approximately 920 days; $72 \times 56 \times 22$ mm; 129 g), and SPOT-375A (after August 2021, lifespan: 915 days; $99 \times 55 \times 21$ mm; 152 g). The Argos transmitter has been widely used for satellite tracking of sea turtles until recently [15, 16, 22, 44], although it could overestimate space use of sea turtles compared to more accurate approaches such as Fastloc-GPS [45, 46]. We attached transmitters to the carapaces of each sea turtle using Aqua Mend Epoxy (PSI-Polymer Systems, NC, USA) and instant mix epoxy (Loctite, CT, USA). Before attaching transmitters, we cleaned the carapace

**Table 1. Physical characteristics and tracking details of the tracked loggerhead and green turtles tracked.**

| ID | CCL (cm) | BW (kg) | Tracking duration (days) | Distance traveled (km) | | Home range (1,000 km$^2$) | | |
|---|---|---|---|---|---|---|---|---|
| | | | | Total | Daily | MCP | 95 PVC | 50 PVC |
| Loggerhead turtle | | | | | | | | |
| KOR0001 | 74.0 | 55.0 | 18/06/2016–23/07/2016 (35) | 129.8 | 5.8 ± 3.8 | 0.4 | 0.4 | 0.0 |
| KOR0008 | 76.0 | 56.0 | 28/09/2017–24/12/2017 (87) | 2,862.2 | 37.9 ± 37.2 | 417.7 | 15.4 | 1.4 |
| KOR0091 | 79.9 | 52.6 | 29/08/2018–13/10/2018 (45) | 1,055.2 | 24.0 ± 14.1 | 36.2 | 41.5 | 5.7 |
| KOR0092 | 91.3 | 85.2 | 17/10/2018–11/09/2019 (329) | 6,217.0 | 18.8 ± 15.7 | 253.9 | 164.3 | 30.7 |
| KOR0149 | 73.2 | 82.2 | 26/08/2021–12/12/2021 (108) | 1,756.5 | 18.4 ± 11.4 | 50.4 | 85.4 | 15.3 |
| KOR0151 | 73.0 | 77.8 | 26/08/2021–09/10/2022 (409) | 8,964.8 | 22.2 ± 14.4 | 489.9 | 333.5 | 23.2 |
| KOR0155 | 71.0 | 69.7 | 25/08/2022–27/09/2022 (33) | 598.5 | 18.1 ± 9.3 | 20.9 | 30.3 | 5.5 |
| Mean | 76.9 ± 6.9 | 68.4 ± 13.8 | (149.5 ± 154.2) | 3,083.4 ± 3,297.3 | 20.7 ± 9.5 | 181.3 ± 205.4 | 95.8 ± 118.4 | 11.7 ± 11.7 |
| Green turtle | | | | | | | | |
| KOR-1 | 65.6 | 31.8 | 29/10/2015–15/12/2015 (47) | 1,601.5 | 32.8 ± 23.0 | 133.1 | 30.7 | 5.3 |
| KOR-2 | 60.0 | 22.5 | 29/10/2015–29/10/2016 (366) | 1,461.3 | 6.2 ± 11.9 | 41.8 | 15.8 | 6.0 |
| KOR0003 | 95.8 | 103.0 | 12/08/2016–17/10/2016 (66) | 1,291.7 | 25.5 ± 26.0 | 75.1 | 25.1 | 7.1 |
| KOR0004 | 68.0 | 41.4 | 02/09/2016–26/10/2016 (54) | 1,452.3 | 26.3 ± 17.4 | 162.1 | 145.4 | 20.7 |
| KOR0009 | 78.8 | 58.9 | 28/09/2017–28/09/2018 (365) | 6,484.3 | 18.7 ± 17.3 | 221.7 | 140.3 | 14.3 |
| KOR0104 | 77.6 | 69.8 | 07/10/2018–01/05/2019 (206) | 5,203.5 | 27.5 ± 22.1 | 1,056.8 | 246.9 | 4.3 |
| KOR0129 | 77.3 | 70.9 | 28/08/2019–26/10/2019 (59) | 2,412.4 | 43.5 ± 33.0 | 221.2 | 244.8 | 34.2 |
| KOR0148 | 70.4 | 79.2 | 26/08/2021–26/07/2022 (334) | 1,623.0 | 7.5 ± 12.5 | 34.7 | 47.3 | 6.6 |
| Mean | 74.2 ± 10.9 | 59.7 ± 26.7 | (187.2 ± 148.4) | 2,691.3 ± 2,004.0 | 23.5 ± 12.5 | 243.3 ± 336.8 | 112.0 ± 96.6 | 12.3 ± 10.5 |

CCL = curved carapace length; BW = body weight; MCP = minimum convex polygon; PVC = percentage volume contour.

using abrasive sandpaper to remove debris, wiped the carapace with isopropyl alcohol to remove greases [47, 48]. To confirm the attachment stability of the transmitter to sea turtles, individuals were monitored for 24 h in an aquarium before release. The tracking period was defined as the period from the first tracking day following the release date until the last day of transmission, which was determined to be the last day of transmission following a period of two months when no transmissions were received. Tracking data were downloaded from the ARGOS system (CLS America, http://www.clsamerica.com). The accuracy estimation of the ARGOS system was classified into six categories: Location class (LC) 0, with an error radius > 1,500 m; LC 1, with > 1,000 m; LC 2, with > 500 m; LC 3, with > 150 m; and LC A and B, with no estimated accuracy. We only used the higher accuracy data for a given day among the recorded daily points, excluding data in LC A and B. We excluded location data on land without a distinct signal from the transmitter or long-distance travel of over 100 km in a single day, which is impossible for sea turtles to travel [15, 16]. Although LC 0, which has lower location quality, could overestimate home range size, the reduction in the number of location points due to the exclusion of LC 0 obscures the understanding the space use patterns of sea turtles [46]. Because the aim of this study was to focus on space use patterns in little-known areas rather than determining the detailed home range size of sea turtles, we included LC 0 for analysis. Additionally, reassessment of location accuracy is necessary considering studies showing that LC A has higher accuracy than LC 0 [45, 49].

## Travel pattern and home range

We measured the distance that sea turtles traveled as the shortest straight line between two consecutively tracked location points in order to study their travel habits. The daily distance traveled (DDT) was calculated as the distance between the given point and the point on the previous day at the most similar time as possible. In addition, the total distance traveled by each turtle was calculated as the sum of the total distances traveled during the entire tracking period.

To define the home range size of individual sea turtles during the tracking periods, we used the minimum convex polygon (MCP) and 95 and 50 percentage volume contours (PVC) based on kernel density estimation, often used to calculate the home range of sea turtles [14, 16, 41, 50]. The MCP and 95 PVC represent general home range sizes, while the 50 PVC indicates the core area of each individual. The distances traveled and home ranges were calculated, after which tracked turtle travel maps were generated using ArcGIS v. 10.1 (ESRI, Redlands, California, USA).

## Habitat use

We compared the DDT of location data inside and outside 50 PVC, and defined the 50 PVC as a foraging area if the DDT inside was significantly smaller, based on the slow travel speed of sea turtles in their foraging areas [51, 52]. On the other hand, we defined all areas outside 50 PVC and where DDT was not significantly small as transitional areas. We also defined 50 PVC nesting areas for adult turtles, that stayed near areas where nesting was previously recorded during their known breeding period. We identified all location points as either foraging, nesting, or transitional areas.

To understand the environmental characteristics of the habitat used by sea turtles, we collected and analyzed data regarding six environmental variables at each location, that are often used for sea turtle studies [14, 50, 53]: distance to shoreline (DTS, km), bathymetry (m), sea surface temperature (SST, ˚C), sea surface salinity (SSS, ‰), chlorophyll-a (Chla, mg/m$^3$), and particulate organic carbon (POC, mol/m$^3$) (Table 2). Distance to shore and bathymetry

**Table 2. Summary of the six environmental variables measured in this study.**

| Environmental variable | Resolution | Frequency | Source |
|---|---|---|---|
| Distance to shore (km) | | | |
| Bathymetry (m) | ~ 1 km | | GEBCO (https://www.gebco.net/) |
| Sea surface temperature (˚C) | ~ 4 km | 8-day mean | NASA OceanColor Web (https://oceancolor.gsfc.nasa.gov) |
| Sea surface salinity (‰) | ~ 40 km | 8-day mean | EarthData (https://earthdata.nasa.gov) |
| Chlorophyll-a (mg/m$^3$) | ~ 4 km | 8-day mean | NASA OceanColor Web (https://oceancolor.gsfc.nasa.gov) |
| Particulate organic carbon (mol/m$^3$) | ~ 4 km | 8-day mean | NASA OceanColor Web (https://oceancolor.gsfc.nasa.gov) |

represent the topographical characteristics of the location where turtles were present, whereas SST and SSS are related to sea surface conditions at each location. Chlorophyll-a and POC, as indicators of the food source such as seagrass, are related to food abundance [29, 54]. Also, we presented the whole tracking data and following environmental variable changes together on the four longest-tracked sea turtles (two loggerhead and two green turtles) and on the one farthest-traveled green turtle to show the relationship between environmental variables along the travel and habitat use of sea turtles. The six environmental variables were extracted using SeaDAS v. 8.3.0 (https://seadas.gsfc.nasa.gov/) and ArcGIS v. 10.1 (ESRI, Redlands, California, USA). Graphs were created using the ggplot2 package [55] in R v. 4.0.5 software [56].

## Statistical analysis

In order to distinguish between the foraging or nesting area and the transitional area, we used an independent t-test, a one-way ANOVA test, and the Kruskal-Wallis test. First, we conducted the Shapiro-Wilkinson test with DDT value of each sea turtle for normality. When only one 50 PVC was generated in a sea turtle individual, the inside and outside DDT were compared using a t-test. If a sea turtle had multiple 50 PVCs, we conducted one-way ANOVA with the Bonferroni post hoc test or Kruskal-Wallis test with the Dunn post hoc test on the inside and outside of each 50 PVC. We defined the inside of 50 PVC that turtles had significantly shorter DDT compared to the outside as the foraging or nesting area.

Based on the environmental variables obtained for the 50 PVC areas, we identified the preferred foraging environment characteristics of both sea turtle species. Additionally, we used an independent sample t-test to compare the interspecific differences in the variables between different foraging areas and the differences in variables between the foraging and transitional areas of loggerhead and green turtles. We considered it statistically significant when the p-value was under 0.05. Statistical analyses were conducted using SPSS v. 23.0 (IBM Corp., Armonk, NY, USA).

## Results

### Satellite telemetry

During the tracking period, 7 loggerhead turtles (6 adults and 1 subadult), with a mean CCL of 76.9 ± 6.9 cm (results are presented as mean ± standard deviation unless otherwise stated) and BW of 52.6 ± 13.8 kg, were tracked for 149.5 ± 154.2 days, traveling between 25.1˚ N, 123.9˚ E to 39.3˚ N, 135.5˚ E (Table 1). The mean DDT by loggerhead turtles was 20.7 ± 9.5 km, while the total distance traveled was 3,083.4 ± 3,297.3 km (range: 129.8–8,964.8). The mean home range size of loggerhead turtles was 181,342 ± 205,355 km$^2$ by MCP and 838 ± 118,402 km$^2$ by 95 PVC, while the core habitat area was 11,703 ± 11,725 km$^2$ by 50 PVC. The mean values of the six environmental variables in the foraging areas of the loggerhead turtles were 54.3 km of

**Table 3. Mean value ± standard deviation of the six environmental variables measured at the transitional and foraging areas of each sea turtle.**

| ID | Type of area | Distance to shore (km) | Bathymetry (m) | SST (˚C) | SSS (‰) | Chla (mg/m³) | POC (mol/m³) |
|---|---|---|---|---|---|---|---|
| Loggerhead turtle | | | | | | | |
| KOR0001 | T | 3.2 ± 3.3 | 21.2 ± 13.8 | 22.8 ± 1.2 | 31.4 ± 1.7 | 1.76 ± 0.94 | 230.8 ± 76.4 |
| | F | - | - | - | - | - | - |
| KOR0008 | T | 129.4 ± 78.0 | 1,795.4 ± 2,110.3 | 26.4 ±1.7 | 33.5 ± 1.4 | 0.49 ± 0.7 | 112.7 ± 165.4 |
| | F | 2.0 ± 1.3 | 13.6 ± 18.3 | 24.0 ± 1.1 | 34.8 ±1.0 | 0.22 ±0.10 | 77.4 ± 25.0 |
| KOR0091 | T | 43.8 ± 49.1 | 85.6 ± 22.3 | 24.6 ± 1.8 | 32.5 ± 1.4 | 0.77 ± 0.37 | 140.4 ± 41.5 |
| | F | - | - | - | - | - | - |
| KOR0092 | T | 58.6 ± 38.3 | 363.6 ± 549.7 | 21.4 ± 2.6 | 33.1 ± 1.4 | 0.52 ± 0.41 | 113.1 ± 66.8 |
| | F | 83.6 ± 55.5 | 119.3 ± 30.2 | 19.0 ± 2.1 | 34.0 ± 1.3 | 0.49 ± 0.25 | 118.0 ± 46.6 |
| KOR0149 | T | 124.7 ± 74.7 | 65.1 ± 19.8 | 21.4 ±3.4 | 31.2 ± 0.9 | 1.62 ± 0.70 | 232.1 ± 78.0 |
| | F | 89.1 ± 22.9 | 79.5 ± 9.7 | 22.6 ± 2.9 | 31.5 ± 0.8 | 1.18 ± 0.63 | 195.1 ± 79.6 |
| KOR0151 | T | 32.4 ± 30.1 | 304.1 ± 502.3 | 20.0 ± 4.1 | 33.1 ± 2.1 | 0.44 ± 0.44 | 101.6 ± 61.1 |
| | F | 42.4 ± 38.1 | 733.9 ± 557.0 | 22.4 ± 3.2 | 32.5 ± 1.6 | 0.69 ± 0.46 | 163.1 ± 112.4 |
| KOR0155 | T | 85.1 ± 67.7 | 87.8 ± 33.1 | 26.8 ± 2.2 | 31.5 ± 1.1 | 0.64 ± 0.47 | 122.8 ± 53.2 |
| | F | - | - | - | - | - | - |
| Mean | T | 68.2 ± 47.3 | 389.1 ± 633.5 | 23.4 ± 2.7 | 32.3 ± 0.9 | 0.89 ± 0.56 | 150.5 ± 56.5 |
| | F | 54.3 ± 40.6 | 236.6 ± 334.4 | 22.0 ± 2.1 | 33.2 ± 1.5 | 0.64 ± 0.41 | 138.4 ± 51.5 |
| Green turtle | | | | | | | |
| KOR-1 | T | 9.7 ± 13.6 | 58.4 ± 42.8 | 21.1 ± 1.4 | 31.5 ± 3.1 | 1.60 ± 1.89 | 167.1 ± 81.8 |
| | F | - | - | - | - | - | - |
| KOR-2 | T | 6.4 ± 8.2 | 71.3 ± 80.9 | 20.2 ± 1.5 | 33.0 ± 1.1 | 0.95 ± 0.88 | 166.0 ± 103.1 |
| | F | 1.5 ± 1.8 | 23.5 ± 20.1 | 23.9 ± 3.7 | 33.6 ± 1.4 | 0.26 ± 0.11 | 69.1 ± 22.6 |
| KOR0003 | T | 9.0 ± 15.9 | 57.2 ± 47.2 | 26.9 ± 2.2 | 31.6 ± 3.1 | 0.7 ± 0.4 | 124.9 ± 39.5 |
| | F | 1.3 ± 1.2 | 27.9 ± 17.7 | 23.8 ± 1.0 | 31.7 ± 0.4 | 0.63 ± 0.15 | 121.9 ± 27.6 |
| KOR0004 | T | 131.2 ± 93.0 | 189.7 ± 278.0 | 26.6 ± 0.9 | 32.4 ± 1.6 | 0.58 ± 0.34 | 104.8 ± 33.7 |
| | F | - | - | - | - | - | - |
| KOR0009 | T | 58.0 ± 46.0 | 465.1 ± 563.8 | 19.6 ± 3.8 | 33.6 ± 1.3 | 0.68 ± 0.81 | 123.0 ± 97.1 |
| | F | 26.9 ± 27.4 | 84.7 ± 80.9 | 21.7 ± 4.3 | 33.8 ± 1.4 | 0.95 ± 0.74 | 167.6 ± 94.2 |
| KOR0104 | T | 32.2 ± 41.4 | 33.0 ± 27.9 | 23.5 ± 2.2 | 32.9 ± 1.6 | 1.59 ± 1.02 | 211.4 ± 89.0 |
| | F | - | - | - | - | - | - |
| KOR0129 | T | 81.0 ± 67.4 | 52.1 ± 40.1 | 25.2 ± 1.5 | 30.6 ± 2.9 | 1.84 ± 1.08 | 230.9 ±91.5 |
| | F | - | - | - | - | - | - |
| KOR0148 | T | 17.1 ± 21.0 | 191.6 ± 207.2 | 18.5 ± 1.5 | 32.4 ± 1.6 | 0.46 ± 0.24 | 105.3 ± 36.7 |
| | F | 1.5 ± 1.5 | 82.7 ± 156.4 | 21.8 ± 4.1 | 33.4 ± 1.6 | 0.50 ± 0.51 | 111.0 ± 59.9 |
| Mean | T | 43.1 ± 44.5 | 139.8 ± 145.5 | 22.7 ± 3.3 | 32.3 ± 1.0 | 1.06 ± 0.54 | 154.2 ± 48.0 |
| | F | 7.8 ± 12.7 | 54.8 ± 33.6 | 22.8 ± 1.2 | 33.1 ± 1.0 | 0.59 ± 0.29 | 104.6 ± 50.1 |

SST = sea surface temperature; SSS = sea surface salinity; Chla = Chlorophyll-a; POC = particulate organic carbon; T = transitional area; F = foraging area.

DTS, 236.6 m of bathymetry, 22.0˚C of SST, 33.2‰ of SSS, 0.64 mg/m³ of Chla, and 138.4 mol/m³ of POC (Table 3).

During the tracking duration, 8 green turtles (6 adults and 2 subadults), with a mean CCL of 74.2 ± 10.9 cm and BW of 59.7 ± 26.7 kg, were tracked for 187.2 ± 148.4 days, with movements ranging from 15.4˚ N, 106.9˚ E to 37.5˚ N, 131.2˚ E (Table 1). The mean DDT by green sea turtles was 23.5 ± 12.5 km, while the total distance traveled was 2,691.3 ± 2,004.0 km (range: 1,291.7–6,484.3). The mean home range size of green turtles was 243,329 ± 336,756 km² by MCP and 112,031 ± 96,626 km² by 95 PVC, and the core habitat area was

$12{,}323 \pm 10{,}466$ km$^2$ by 50 PVC. The mean values of the environmental variables in the foraging areas of green turtles were 7.8 km of DTS, 54.8 m of bathymetry, 22.8˚C of SST, 33.1‰ of SSS, 0.59 mg/m$^3$ of Chla, and 104.6 mol/m$^3$ of POC (Table 3).

## Seasonal travel pattern

Seven loggerhead turtles were released in South Korea at 33.2˚ N– 34.6˚ N during summer and autumn, from June to September (Table 1). Five loggerhead turtles traveled southward from the release location and stayed below 35˚ N throughout the tracking period, whereas KOR0092 and KOR0151 traveled up to the East Sea at 39˚N during the summer and autumn, between July and early December (Fig 1). Eight green turtles were released in South Korea at 33.2˚ N– 35.2˚ N during summer and autumn, from August to October (Table 1). Similar to the loggerhead turtles, seven green turtles moved southward from the release location and stayed below 35˚ N. Only KOR0009 traveled to the East Sea up to 37.5˚ N during the spring and summer, between April and July (Fig 1).

## Habitat use and environmental variables

Loggerhead turtles used areas in the sea near Okinawa Island (KOR0008), Tsushima Islands (KOR0092), the East China Sea close to Jeju Island (KOR0091 and KOR0092), and the east coastal sea of the Gangwon Province in South Korea (KOR0151) for foraging (Fig 2 and S2 Table). No loggerhead turtles were presumed to have used the nesting areas during the tracking period because the areas they used had no previous nesting records. Okinawa Island, which is the only known nesting area, was only used during the non-breeding season. In foraging areas, loggerhead turtles traveled shorter distance per day (p-value < 0.05) and stayed closer to the shore than during transitional periods (p-value < 0.001). In addition, the foraging sites of loggerhead turtles had lower SST (p-value < 0.05), higher SSS (p-value < 0.05), and lower POC (p-value < 0.05) values than of transitional areas (Table 3).

The major foraging areas for four green turtles were identified as follows. KOR-2 used the sea close to Tanega Island, while KOR0003 used the sea close to Jeju Island. KOR0148 used the sea near Jeju Island in South Korea and Uji Island, and KOR0009 used three foraging areas in the sea near Yeosu and Ulsan, and the East China Sea, west of Kyushu in Japan (Fig 3 and S2 Table). Furthermore, the nesting area of KOR0104 was located near Hainan Island, a historical nesting site for green turtles [57, 58]. The DDT (p-value < 0.05) and DTS (p-value <0.05) of green turtles were larger in their transitional areas than those in their foraging areas (Table 3). Considering the bathymetry values, green turtles used shallower areas for foraging than that used for travel (p-value < 0.05). There was no significant difference in SST between the transitional and foraging areas (p-value < 0.05), whereas SSS was higher in transitional areas than that in foraging areas (p-value < 0.05). Both Chla (p-value < 0.05) and POC (p-value < 0.05) were higher during travel than during foraging.

In the foraging areas used by loggerhead (n = 408) and green turtles (n = 545), the observed variables corresponding to loggerhead turtles showed a longer mean DDT (p-value < 0.05) and larger DTS (p-value < 0.05), deeper bathymetry (p-value < 0.05), lower SST (p-value < 0.05), lower SSS (p-value < 0.05), and higher Chla (p-value < 0.05) and POC (p-value < 0.05; Table 4) than those corresponding to green turtles (Table 3). Also, we represented the monthly changes in foraging areas of the six variables for both sea turtles (Fig 4).

## Representative habitat use pattern

KOR0092 (329 days) and KOR0151 (401 days), which were the longest-tracked loggerhead turtles, gave us the most detailed information about the environment in the foraging area (Fig

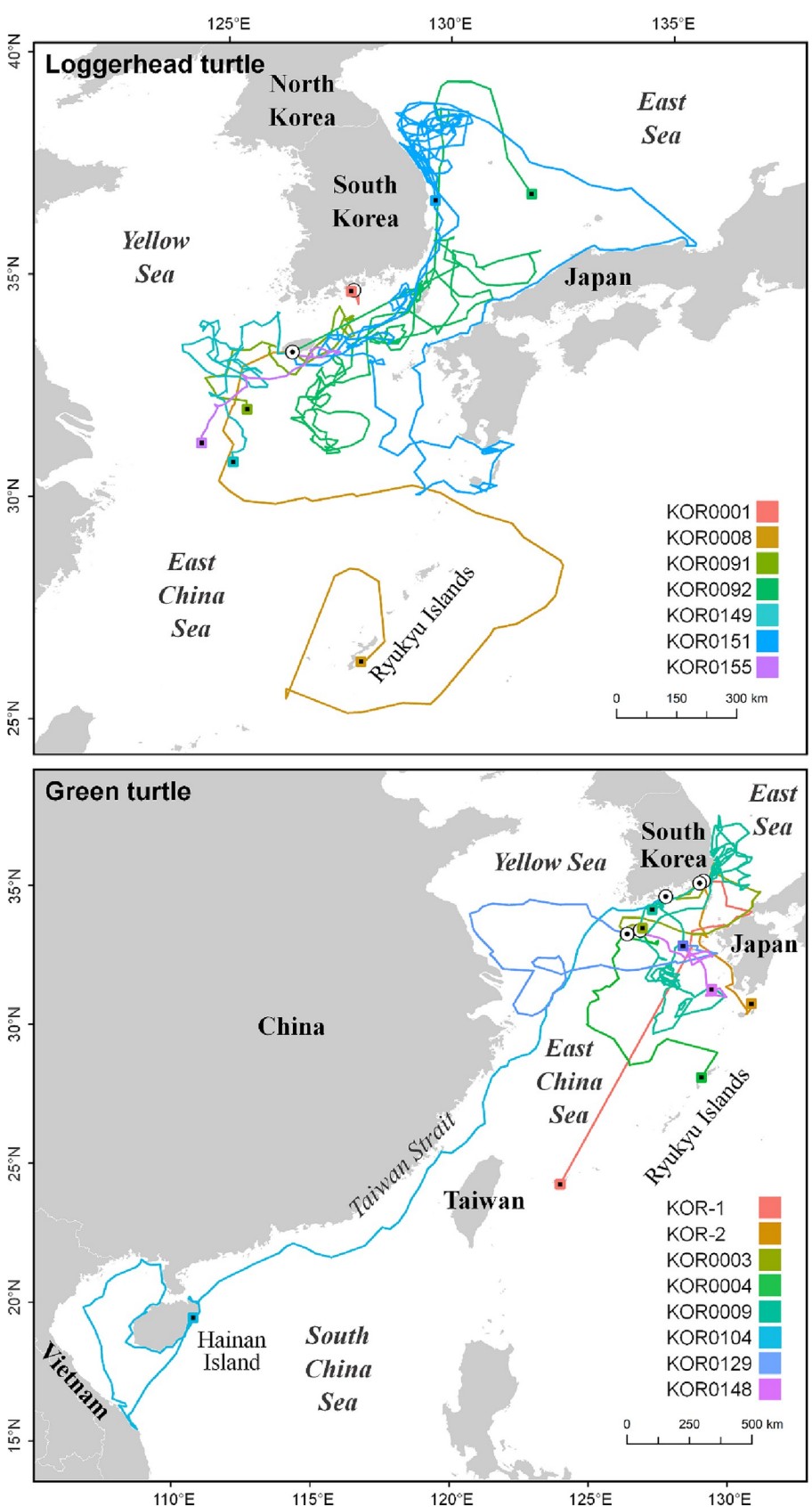

**Fig 1. Summarized travel patterns of the sea turtles released at the coastal seas of South Korea.** White circles and squares with each color indicate start and end points, respectively. This map was generated based on GADM data (https://gadm.org/).

5). Loggerhead turtle KOR0092 first stayed in the sea south of Jeju Island, between December 10, 2018 and April 17, 2019. After release, KOR0092 moved to and stayed at the sea near Tsushima Island, from April 23, 2019 to May 23, 2019, and from June 7, 2019 to July 22, 2019. Its first major foraging area was located in the open East China Sea, more than 100 km away from land during winter, and with an average of 0.54 mg/m$^3$ of Chla and 129.2 mol/m$^3$ of POC. The

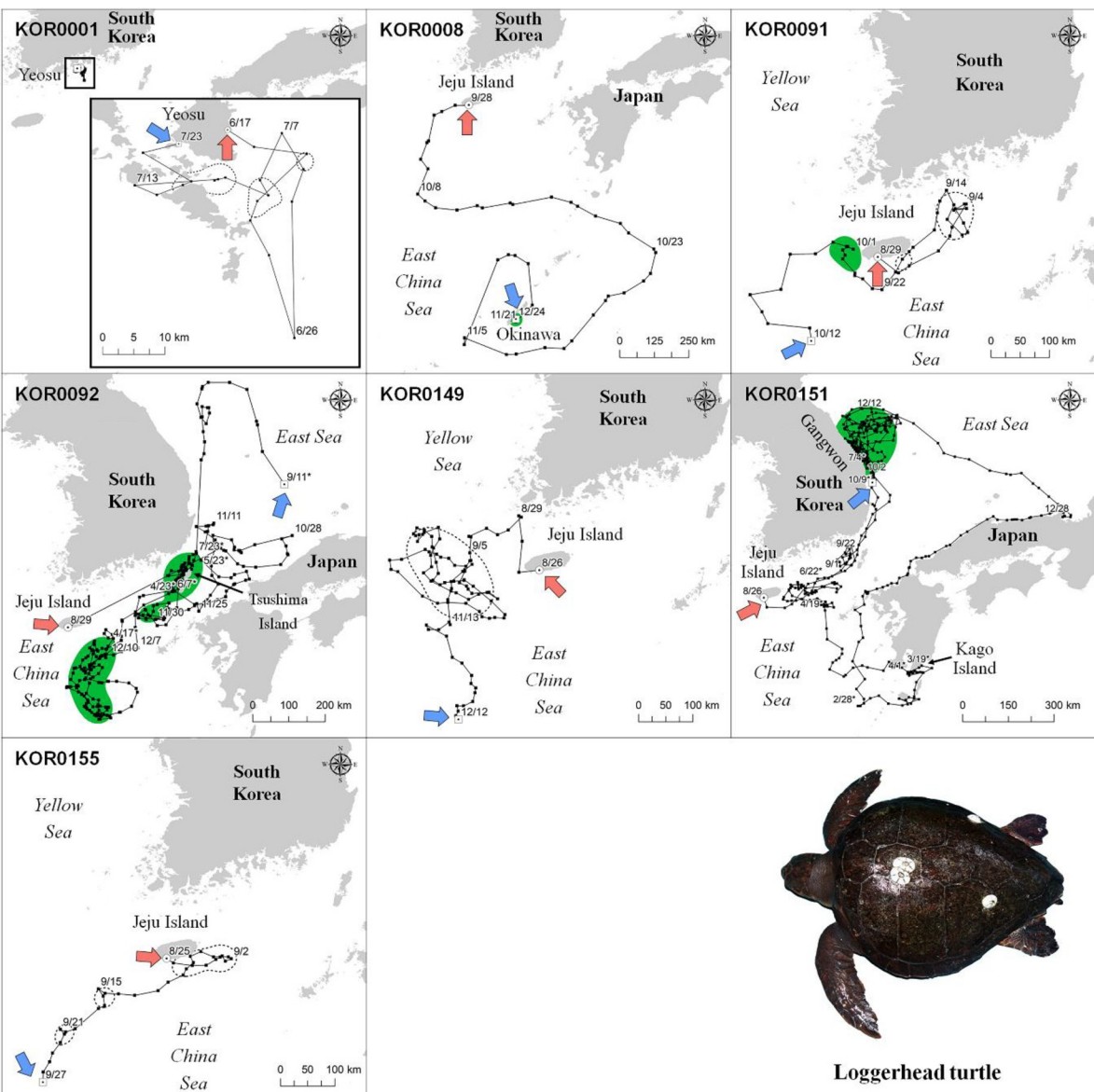

**Fig 2. Travel patterns and core habitat of the seven satellite-tracked loggerhead turtles.** Green zones indicate foraging areas calculated by 50 PVC. Dotted line areas indicate 50 PVC zones that were excluded from the core habitat. Dates marked with asterisks correspond to the year following the year of release. The red and blue arrows indicate the start and end dates of the tracking period, respectively. This map was generated based on GADM data (https://gadm.org/).

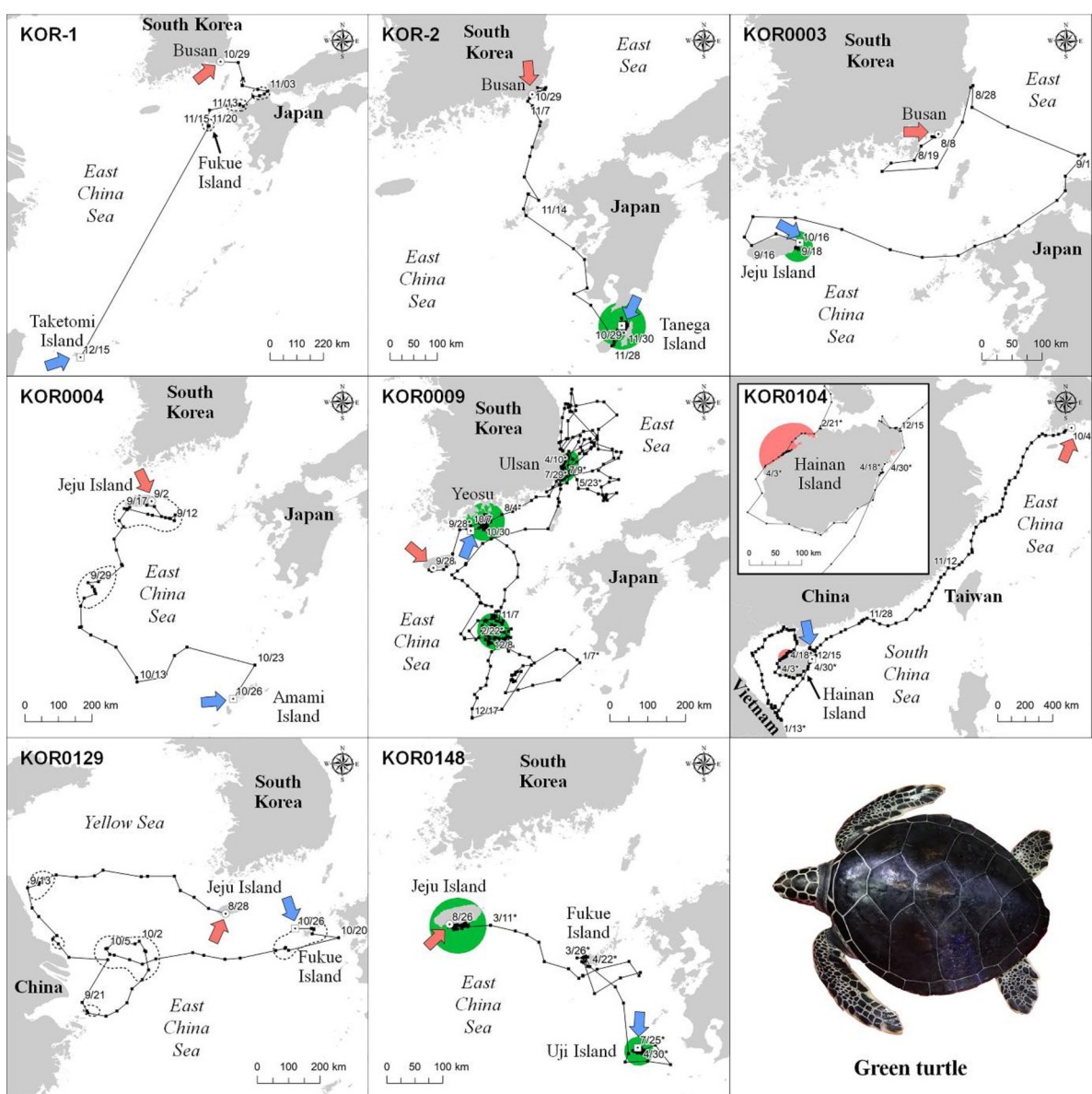

**Fig 3. Travel patterns and core habitat of the eight satellite-tracked green turtles.** Green and red zones indicate foraging and nesting areas calculated by 50 PVC, respectively. Dotted line areas indicate 50 PVC zones that were excluded from the core habitat. Dates marked with asterisks correspond to the year following the year of release. The red and blue arrows indicate the start and end dates of the tracking period, respectively. This map was generated based on GADM data (https://gadm.org/).

second major foraging area of KOR0092 was located near Tsushima Island, mainly during the spring and summer (Fig 5). Immediately after its release, KOR0151 moved to the coast sea east of the Gangwon Province between autumn and early winter. At the start of winter, KOR0151 traveled southward toward Kago Island and used the sea in southern Japan until the end of winter. KOR0151 stayed near the eastern sea of Jeju Island for two months in spring, after which it returned to the eastern coast sea of the Gangwon Province and used it for three months from summer until the end of tracking in autumn.

Compared to loggerhead turtles, green turtles showed wider travel ranges in the sea around South Korea, Japan, China, and Vietnam (Fig 1). Among the eight green turtles, KOR0009 was

**Table 4. Comparison of the six environmental variables within and between species.** Interspecific differences were only compared in foraging areas corresponding to both species, while the columns with each species title indicate differences in the variables between the foraging and travel areas using t-test. Asterisk indicates significant differences (p-value < 0.05).

| Variables | Interspecific foraging use | Loggerhead turtle | Green turtles |
|---|---|---|---|
| DDT | t = 15.314 | t = 6.997 | t = 21.294 |
| | $p < 0.001$* | $p < 0.01$* | $p < 0.001$* |
| DTS | t = 24.622 | t = -2.104 | t = 16.249 |
| | $p < 0.001$* | $p < 0.05$* | $p < 0.001$* |
| Bathy | t = -14.767 | t = -0.587 | t = -8.840 |
| | $p < 0.001$* | $p > 0.05$ | $p < 0.001$* |
| SST | t = -6.281 | t = 2.878 | t = -1.516 |
| | $p < 0.001$* | $p < 0.01$* | $p > 0.05$ |
| SSS | t = -4.208 | t = -2.718 | t = -8.013 |
| | $p < 0.001$* | $p < 0.01$* | $p < 0.001$* |
| Chla | t = 3.300 | t = -0.210 | t = 11.307 |
| | $p = 0.001$* | $p > 0.05$ | $p < 0.001$* |
| POC | t = 6.411 | t = - 3.479 | t = 9.751 |
| | $p < 0.001$* | $p < 0.01$* | $p < 0.001$* |

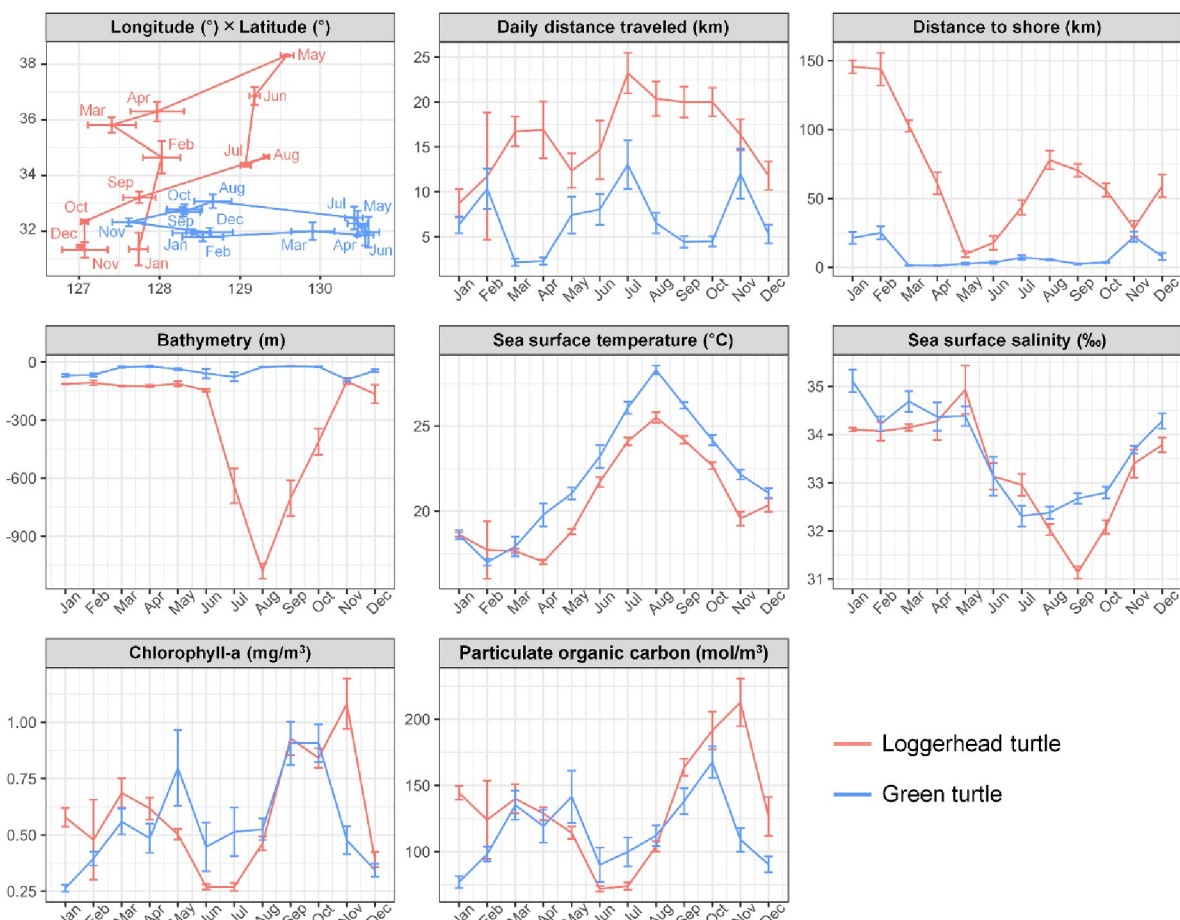

**Fig 4. Monthly changes in latitude, daily distance traveled, and six environmental variables (mean ± standard error) measured in the foraging areas of the both sea turtle species.**

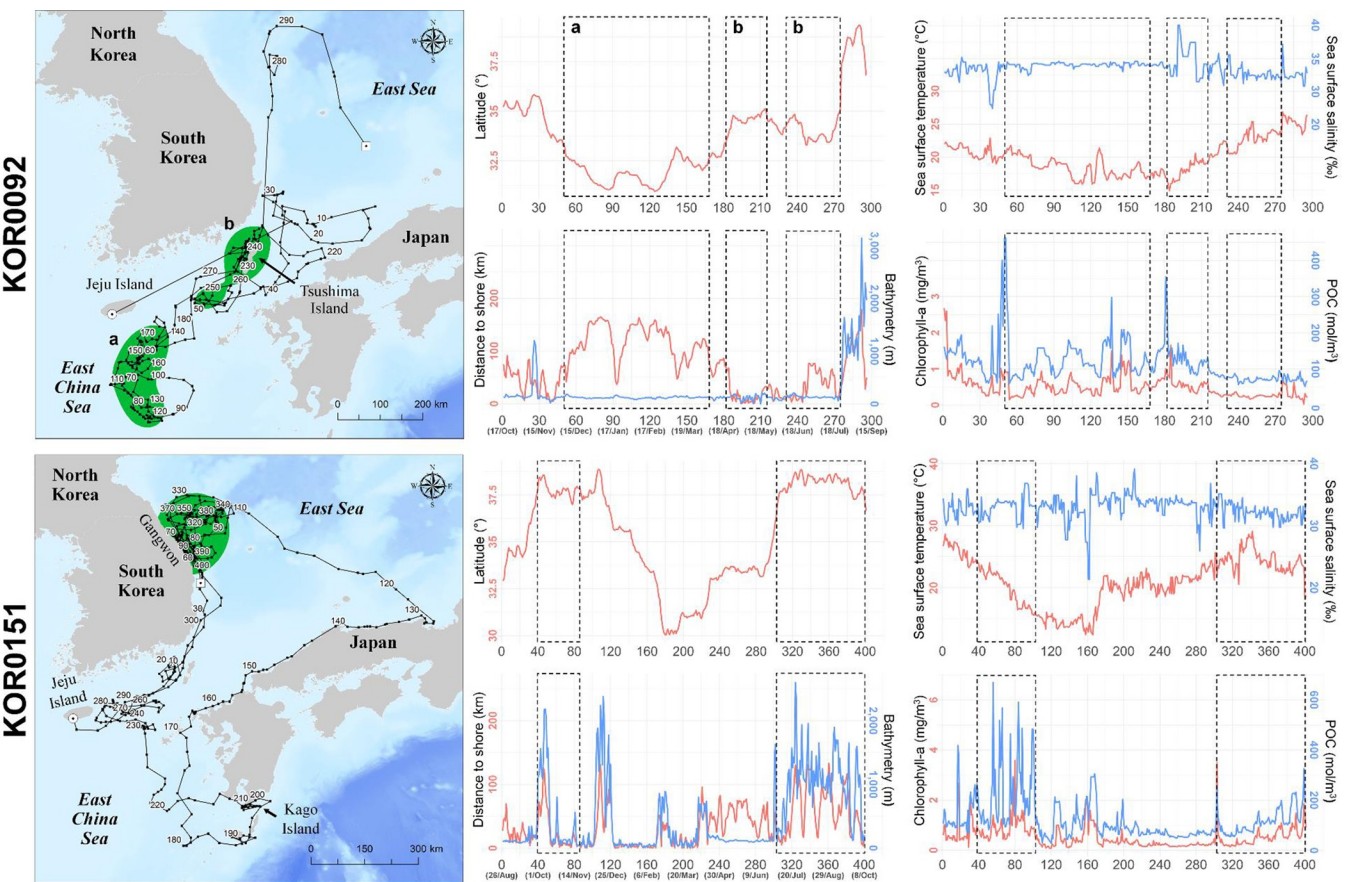

**Fig 5. Travel routes and environmental variables on the travel route of two representative loggerhead turtles.** Green zones on the maps and the dotted sections in the graphs indicate core areas by 50 PVC. POC = particulate organic carbon. This map was generated based on GADM data (https://gadm.org/).

tracked for the longest period (365 days) and showed three seasonal habitats. Green turtle KOR0009 used three core areas: the coastal sea of Yeosu in South Korea, in autumn (from October 7, 2017 to October 30, 2017 and later from August 4, 2018 to September 28, 2018), the East China Sea in winter (from November 7, 2017 to December 8, 2017 and later from January 15, 2018 to February 22, 2018), and the coastal sea near Ulsan, Gyeongnam Province in South Korea, in spring and summer (from March 24, 2018 to April 10, 2018, from May 24, 2018 to June 5, 2018, and from July 9, 2018 to July 25, 2018). After its release at Jeju Island, KOR0009 used the coastal sea near Yeosu in autumn and traveled to the East China Sea, located at the lowest latitude for overwintering and where the Chla (0.28 mg/m$^3$) and POC (76.9 mol/m$^3$) were the lowest. After winter, KOR0009 traveled to the coastal sea of Ulsan, which had 1.26 mg/m$^3$ of Chla and 217.8 mol/m$^3$ of POC, where it stayed between spring and summer. Finally, KOR0009 returned to the coastal sea of Yeosu in autumn, where it stayed until the end of the tracking.

KOR0104 was tracked for 206 days and showed a distinct travel pattern reaching Vietnam and Hainan Island, which represents the farthest distance traveled during this study (Fig 6). Upon release in Yeosu, KOR0104 continuously traveled to the South Central Coast of Vietnam for 100 days until January 13, 2019, after which it traveled backward and arrived at Hainan Island, China. The core habitat of KOR0104 was located near the northwest coast of Hainan Island, where it stayed from February 26, 2019, to April 3, 2019. Its core habitat had 7.2 m of

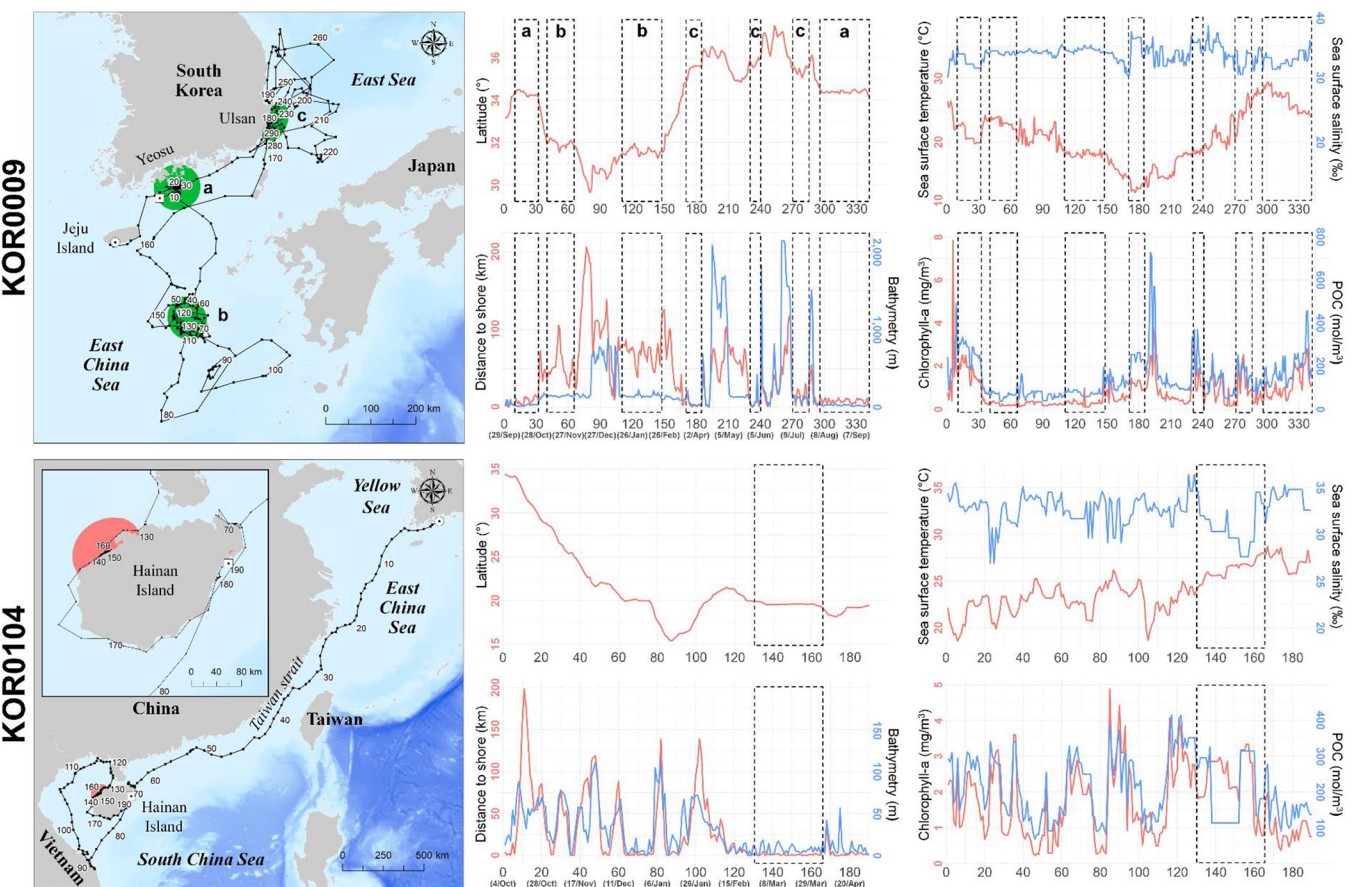

**Fig 6. Travel routes and environmental variables on the travel route of two representative green turtles.** Green and pink zones on the maps and the dotted sections in the graphs indicate core areas by 50 PVC. POC = particulate organic carbon. This map was generated based on GADM data (https://gadm.org/).

bathymetry, 26°C of SST, 2.30 mg/m$^3$ of Chla, and 202.0 mol/m$^3$ of POC, and an average distance from the shoreline of 1.2 km (Fig 6).

## Discussion

### Habitat use of loggerhead turtle

In this study, loggerhead turtles used the open East China Sea, the coastal sea adjacent to the southern Korean Peninsula and western Japan, and the East (Japan) Sea as their main foraging areas, including the coastal seas of Okinawa Island, Jeju Island, Tsushima Island, and Gangwon Province. Japan has the largest nesting area for loggerhead turtles in the North Pacific Ocean. Major foraging areas adult turtles are located in the seas around Japan, including the East China Sea, the sea west of the Ryukyu Islands, the open North Pacific Ocean, and the southern coastal area of the Japanese Archipelago [24, 59]. However, in the present study, the seven tracked loggerhead turtles mainly used the northern part of the East China Sea as foraging areas, which have higher latitudes than those of previously reported areas, such as those near Jeju Island, Tsushima Island, and the east coast of South Korea. These results indicate that the foraging range of loggerhead turtles in the western Northwest Pacific Ocean is wider than previously reported.

All seven loggerhead turtles mainly used the northern part of the East China Sea, which is a suitable habitat for the species owing to being shallow, with a sea depth of less than 100 m and

having an annual SST over 20°C [5, 60]. Additionally, in this region, small islands such as Jeju Island, Tsushima Island, and Okinawa Island provide core foraging areas for loggerhead turtles [5, 22, 61, 62]. Furthermore, despite the few records of loggerhead turtles in the East Sea, KOR0092 and KOR0151 periodically used the East Sea in this study. These two turtles showed a foraging loop behavior, using the East Sea in summer and autumn when the SST was higher and the southern East China Sea in winter when the SST was lower. KOR0092 and KOR0151 traveled south in late November and early December and back to the East Sea in the summer. Despite the East Sea having a depth of less than 1,000 m and a relatively low SST, its Chla and POC contents were high, possibly because of an abundance of food sources. Considering such environmental conditions, KOR0092 and KOR0151 might consume enough food in the East Sea during the warm summer and early autumn seasons and then travel to the southern East China Sea for overwintering in the cold winter when the SST drops. Loggerhead turtles use the East Sea near the eastern shoreline of South Korea to forage various seaweeds, jellyfish, and shellfish [63]. In a previous report, over 90% of the loggerhead turtles studied near the Korean Peninsula, with a higher latitude and a lower SST than those of southern Japan, were observed between summer and autumn, between June and November [34]. The main reason for the observed pattern is thought to be related to low sea temperatures. The coastal sea of South Korea has an SST below 15°C from December to May [64]; as loggerhead turtles are vulnerable at this sea temperature, they are rarely observed [22, 36, 65]. Currently, with the sea temperature of the Pacific Ocean continuously rising largely due to climate change, the amount of food sources in the traditional foraging areas of loggerhead turtles is decreasing, forcing them to exploit cooler foraging areas that have more available food sources [9]. As sea temperatures in the Northwest Pacific Ocean increase further, more loggerhead turtles can be expected to stay longer in the East Sea and use it as a foraging area.

## Habitat use of green turtles

Except for KOR0104, which traveled to Hainan Island, green turtles mainly used the East China Sea between South Korea and Japan, the coastal sea of the Korean Peninsula, and small islands in southern Japan as foraging areas. The major foraging areas of green turtles in the Northwest Pacific Ocean are the coastal sea of southern Japan and the Ryukyu Islands [37]; however, in this study, among the eight turtles, only KOR-1 and KOR0004 traveled to the Ryukyu Islands. The remaining five green turtles, except for KOR0104, used the northern East China Sea as their major foraging area. Tanega Island is a traditional foraging or nesting area for green turtles [60], which was used in the present study by the subadult KOR-2 as a foraging area. Jeju Island, which is also a major foraging area for green turtles [34, 41], was used by two green turtles, KOR0003 and KOR0148, as their major foraging area in this study. Moreover, they seemed to also use it as an overwintering area considering they stayed there in autumn and winter. Comparable to our results, five of eight green turtles studied by Jang et al. (2018) mainly stayed near Jeju Island during the tracking period, including during winter [41]. These results suggest that green turtles use the upper East China Sea, including the seas near Jeju Island and the southern Korean Peninsula, as both foraging and overwintering areas.

Adult green turtles can have high habitat fidelity for more than one foraging area [4, 26, 66]. In particular, KOR0009 showed typical foraging loop behavior in three seasonal habitats. Immediately after release, KOR0009 stayed in the sea neighboring of Yeosu in South Korea, in autumn, used the East China Sea at the lowest latitude during winter, used the sea around Ulsan, Gyeongnam Province in South Korea, at the highest latitude during summer, and returned to Yeosu in autumn. Rather than the loggerhead turtles, green turtles have also been reported to use the East Sea near the eastern shoreline of the Korean Peninsula mainly to

forage on seaweeds [63]. Among the three habitats, food-related Chla and POC were higher in the sea near Yeosu and Ulsan at high latitudes and relatively low in the East China Sea. Therefore, like loggerhead turtles KOR0092 and KOR0151, green turtle KOR0009 also remained in the coastal area of South Korea, which provides abundant food sources despite its relatively low sea temperatures, and traveled to the southern East China Sea before winter, when its thermoregulation is limited. Additionally, green turtles consume animal matter more than seagrasses at high latitudes or low SST areas under 25˚C [67]. However, although green turtles generally used areas with a SST below 23˚C in our study, the most abundant food item was seaweed according to previous research about stomach contents [34]. This indicates that green turtles use waters off South Korea as foraging areas mainly in the warm season and travel to lower latitudes in the cold season.

KOR0104 was the only turtle that was presumed to have traveled to a nesting area, even though its destination, Hainan Island, has had no records of green turtle nesting in the past 37 years owing to coastal development [57, 58]. Nevertheless, KOR0104 visited 12 of the 13 historical nesting areas on Hainan Island [58] and appeared to aim to nest regardless of its success. The Yaeyama and Ogasawara Islands, which are closer to the release site, are the largest nesting sites for green turtles in the Northwest Pacific Ocean [68], so the long travel of KOR0104 to Hainan Island suggests that this turtle may have originated in the southwestern Pacific. Previous genetic analyses using mtDNA have shown that green turtles in the Northwest Pacific Ocean are intermixed with turtles from Southeast Asian colonies [19, 20, 68]. Other studies have also reported the Taiwan Strait being used by other green turtles for travel [14, 69]. Therefore, this route could be a major passage for green turtles, connecting the northern and southern parts of the western Pacific Ocean.

## Comparison of the two sea turtle species

Both the loggerhead and green turtles mainly stayed in the East China Sea, while a few individuals used the East Sea, mainly in summer and autumn. Green turtles showed a wider distribution and more diverse travel patterns than loggerhead turtles, with one green turtle travelling to the South China Sea. Additionally, differences in the environmental variables between the foraging habitats of the two species were identified. Throughout the year, green turtles stayed in foraging areas within 26 km of the shoreline and used the areas with depths of less than 100 m, while loggerhead turtles used areas with depths of approximately 150 m from winter to spring and over 1,000 m in summer and autumn in the East Sea. In addition, loggerhead turtles stayed approximately 50 km away from the shoreline during winter and summer. Both adult loggerhead and green turtles are known to forage mainly in shallow coastal waters, where the depths reach between 20 and 50 m [50, 66], although some turtles prefer open oceanic habitats [4, 70]. In Korean waters, more loggerhead turtles are observed in the East Sea, whereas more green turtles are observed near Jeju Island [5, 34, 41]. Because two loggerhead turtles, KOR0092 and KOR0151, stayed in the East Sea for long periods, the values of Chla and POC in the foraging areas of loggerhead turtles were greater than those of green turtles. These interspecific differences suggest that loggerhead turtles are relatively better adapted to deeper waters and cooler sea temperatures than green turtles [52].

## Conservation strategy

The information obtained through the tracked sea turtles expands our horizons on the travel patterns and habitat use of turtles in the Northwest Pacific Ocean, where little is known about the spatial ecology of sea turtles. The home range of sea turtles could be inaccurate given the status of the studied animals as rescued. Nevertheless, the periodic behavior of tracked sea

turtles and their response to environmental variables, especially SST, Chla, and POC, showed that the rescued turtles were rehabilitated well and are re-adapting to the sea. Our results revealed patterns of habitat use different from those reported in previous results [24, 37, 59]. Contrary to what has been previously reported, adult sea turtles did not migrate to the open Central Pacific Ocean in our study [19, 41], and instead stayed in and used the East China Sea and East Sea around South Korea and Japan as their main foraging areas. Notably, this study marks the first time the East Sea (east coast of the Gangwon Province in South Korea) has been identified as a sea turtle foraging area. The low sea temperatures in the northern part of the East China Sea and East Sea have been reported to limit the habitat use of loggerhead and green sea turtles [34, 37]. However, recent climate change causes a continuous increase in sea temperatures in the North Pacific Ocean, which could extend northward into the habitat [9, 10, 71]. During the past 50 years, the sea surface temperature around South Korea has increased by approximately 1.5°C, which is near three times greater than the global average increase [72, 73]. Although some sea turtles were observed using the East Sea during warm summers and autumns in this study, the number of sea turtles that do so is expected to rapidly increase in following years due to climate change. The appearance of sea turtles in the coastal sea of South Korea and the steady increase in the number of fishery bycatch records over the last 15 years [34, 38, 39] support this prediction. Therefore, it is necessary to continuously update the occurrence pattern of sea turtles in these areas and identify potential problems that sea turtles will face following population increases. Given that fishery bycatch is responsible for a global decline of sea turtles [7, 74, 75 rid="_Ref162590133"], protection policies for sea turtles and monitoring plans for fishing vessels should be prepared and applied at a national level. In conclusion, the increased interest in and implementation of management policies for sea turtle habitats in the northern part of the East China Sea and the East Sea near the Korean Peninsula and Japan could play a critical role in the conservation of sea turtles in the near future.

## Supporting information

**S1 Table. Rescue and release data of tracked sea turtles.**
(DOCX)

**S2 Table. Comparison of the daily distance traveled (km/d) by sea turtles between 50 PVCs and outer areas.**
(DOCX)

**S1 File. Sea turtle movebank file.**
(ZIP)

## Acknowledgments

We thank Min-Woo Park and Hyerim Kwon for their assistance with data management.

## Author Contributions

**Conceptualization:** Il-Hun Kim, Il-Kook Park.

**Data curation:** Il-Hun Kim, Il-Kook Park.

**Formal analysis:** Il-Kook Park, Daesik Park.

**Funding acquisition:** Min-Seop Kim, In-Young Cho, Won Joon Shim, Yong-Rock An.

**Investigation:** Il-Hun Kim, Min-Seop Kim, In-Young Cho, Dongwoo Yang, Dong-Jin Han, Eunvit Cho, Won Joon Shim, Sang Hee Hong.

**Methodology:** Daesik Park.

**Project administration:** Yong-Rock An.

**Resources:** Il-Hun Kim.

**Software:** Il-Kook Park.

**Supervision:** Daesik Park, Yong-Rock An.

**Validation:** Daesik Park.

**Writing – original draft:** Il-Hun Kim, Il-Kook Park, Daesik Park.

**Writing – review & editing:** Il-Hun Kim, Il-Kook Park, Daesik Park, Yong-Rock An.

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
