## [Decision Letter · Decision Letter 0]

11 Sep 2023

PONE-D-23-24778Habitat use of loggerhead (Caretta caretta) and green (Chelonia mydas) turtles in the Northwest Pacific Ocean, the northern limit of their distribution rangePLOS ONE

Dear Dr. An,

Thank you for submitting your manuscript to PLOS ONE. I have now received reviews from two experts in the field of animal tracking, including sea turtles, who have graciously self-identified themselves as part of the review process, Graeme Hays (Reviewer 1), and Jeff Seminoff (Reviewer 2). Both reviewers concluded that you have presented a valuable data set, but they also both make a number of constructive suggestions that should improve the impact and value of your study to the larger scientific community. After careful consideration, I would very much appreciate if you would revise the manuscript and take into account the recommendations made by the reviewers, including suggestions that will improve the analysis and interpretation of your telemetry data. 

I look forward to receiving your revised manuscript.

Kind regards,

Lee W Cooper, Ph.D.

Section Editor

PLOS ONE

Journal Requirements:

"This study was supported by grants from the National Marine Biodiversity Institute of Korea (2023M00300 and 2023E00300) and the ‘Development of Technology for Impact Assessment of Marine Plastic Debris on Marine Ecosystem (EA0114)" funded by the Korea Institute of Ocean Science and Technology."

5. We note that [Figures 1,2,3,5, and 6] in your submission contain [map/satellite] images which may be copyrighted. All PLOS content is published under the Creative Commons Attribution License (CC BY 4.0), which means that the manuscript, images, and Supporting Information files will be freely available online, and any third party is permitted to access, download, copy, distribute, and use these materials in any way, even commercially, with proper attribution. For these reasons, we cannot publish previously copyrighted maps or satellite images created using proprietary data, such as Google software (Google Maps, Street View, and Earth). For more information, see our copyright guidelines: http://journals.plos.org/plosone/s/licenses-and-copyright.

a. You may seek permission from the original copyright holder of Figures 1,2,3,5, and 6 to publish the content specifically under the CC BY 4.0 license.  

Reviewers' comments:

Reviewer's Responses to Questions

**Comments to the Author**

1. Is the manuscript technically sound, and do the data support the conclusions?

Reviewer #1: Partly

Reviewer #2: Partly

2. Has the statistical analysis been performed appropriately and rigorously? 

Reviewer #1: Yes

Reviewer #2: No

3. Have the authors made all data underlying the findings in their manuscript fully available?

Reviewer #1: No

Reviewer #2: No

4. Is the manuscript presented in an intelligible fashion and written in standard English?

Reviewer #1: Yes

Reviewer #2: Yes

5. Review Comments to the Author

Reviewer #1: Here Argos satellite tags are deployed on rehabilitated turtles and then their movements are described. I congratulate the authors on obtaining such nice tracking data. This is a solid, descriptive account of where tracked turtles moved. The writing is a little parochial, with few wider issues of general interest being considered. There are also methodological issues with the interpretation of the tracking data. With careful revision I think this will be a solid contribution.

1 Line 42. I think you need to be a little more honest and say that in many area turtle numbers are increasing, thanks to conservation efforts and so in the latest IUCN regional assessments not all populations are endangered. E.g. see: Mazaris AD et al. (2017). Global sea turtle conservation successes. Science Advances 3: e1600730. https://doi.org/10.1126/sciadv.1600730

If you look on the IUCN website at the latest regional assessments, you will see that in some areas turtles are not listed as endangered.

2. Line 49. “expanding north”. I think you mean “polewards”. In the southern hemisphere species are heading south.

3. Line 51. “telemetry is a useful method for investigating the spatial ecology and habitat use of sea turtles …” I would simply cite a recent review: “Satellite tracking sea turtles: opportunities and challenges to address key questions. https://doi.org/10.3389/fmars.2018.00432”

4. Lines 59. “Overall, the habitat information and environmental characteristics identified by satellite telemetry could contribute to the effective protection and restoration of sea turtle populations …”

Perhaps the best synthesis of how satellite tracking has helped drive turtle conservation is within this review: “Translating marine animal tracking data into conservation policy and management. Trends in Ecology and Evolution 34, 459-473. https://doi.org/10.1016/j.tree.2019.01.009”

5. Line 86. “Considering the minimum CCL of sexually mature sea turtles is known as 73 cm in loggerhead turtles and 67 cm in green turtles …”

One of your green turtles was 96cm. I find it hard to believe there could be such a massive range of adult sizes from 67-96cm. I suggest you discuss whether any of these turtles were actually sub-adults.

6. Normally green turtles feed on seagrass or macroalgae. There are recent reviews on their diet (https://doi.org/10.1007/s00227-020-03786-8). In this context I would discuss how many of the tracked turtles foraged in shallow coastal areas versus being in deep oceanic areas.

7. Table 3. You talk about “travelling” and “foraging”. How did you distinguish these 2 different states ?

8. Lines 102. “… classes A and B, with no estimated accuracy”.

This is not true.

Class A and B accuracy has been assessed in various studies. See for example:

Witt MJ et al. (2010). Assessing accuracy and utility of satellite-tracking data using Argos-linked Fastloc-GPS. Animal Behaviour 80, 571-581. doi:10.1016/j.anbehav.2010.05.022

Hays GC et al. (2001). The implications of location accuracy for the interpretation of satellite tracking data. Animal Behaviour 61, 1035-1040.

These studies show that Class A locations are good.

The worst accuracy is class 0 locations.

So I think you need to omit class 0 locations and include class A locations.

Also note that now Argos can provide Kalman filtered locations. The users decides. So specify whether you used Kalman filtered locations or not.

9. Line 113. “ … we used the minimum convex polygon (MCP) and 95 and 50 percent volume contours (PVC) based on kernel density estimation.

You need to discuss that you likely over-estimate home-range using low quality Argos tracking, compared to if you has used high-resolution Fastloc GPS tracking. See:

Thomson JA et al. (2017). Implications of location accuracy and data volume for home range estimation and fine-scale movement analysis: comparing Argos and Fastloc-GPS tracking data. Marine Biology 164, 204. DOI 10.1007/s00227-017-3225-7

This is not necessarily a problem but you need to acknowledge this issue in the discussion with words like: “We appreciate that we likely overestimated space use by turtles by using Argos locations which have low accuracy compared to other approaches such as Fastloc-GPS (ref)”.

10. The discussion is rather parochial which will limit readership. I would try and consider some wider issues. I think you need to discuss whether these turtles were behaving normally. Usually adult turtles travel back to the same foraging site to which them maintain fidelity. See:

Shimada T et al. (2020). Fidelity to foraging sites after long migrations. Journal of Animal Ecology 89, 1008–1016. https://doi.org/10.1111/1365-2656.13157

Other have tracked rehabilitated turtles. Are they behaving normally ?

11. Linked to point 7, I would specify where turtles were originally captured and how far away these sites were from the release points.

12. I was asked to comment on the data availability. It is not clear to me where the raw tracking data can be accessed (time stamps lat/longs for each turtle). I suggest you deposit in Movebank or a similar data bank.

In summary, a solid, descriptive account of the movements of tagged turtles. Graeme Hays

Reviewer #2: Thank you for the opportunity to review "Habitat use of loggerhead (Caretta caretta) and green (Chelonia mydas) turtles in the Northwest Pacific Ocean, the northern limit of their distribution range." This is an important study theme and having more telemetry data for the region will be useful for expanding our knowledge of the biology of both species, and also provide information for conservation action.

In reading the paper, I had started to do line-by-line suggested edits (see below), but I only provide such edits until the Methods section. This is because as I was reading the paper, I discovered that had several fundamental comments and concerns about the paper, and thus this review has largely focused on conceptual themes relating to movement analysis and home range estimation.

First, for loggerheads, the statement that this is at the northern limit of their distributional range should always be clarified that you are talking about the waters of the western Pacific near Japan. Indeed, 39.3N is a northern limit for green turtles, and loggerheads near the Japanese prefecture, but ss per Kobayashi et al. (2008, JEMBE) and others, loggerheads in the Central North Pacific swim to higher than 45N. Thus the title of this paper is confusing needs to be slightly adjusted after the ‘,’ to make it clear you are referring to the NWP only. The same is true for all other mentions of this throughout the paper.

Second, I have major concerns about the authors' application of the term Home Range. This term is usually applied to movement data for turtles that set up residency in localized areas and remain inside clear patial boundaries for long periods of time (e.g. green turtle benthic foraging in coastal habitats). However, for animals living in offshore waters, home range is not the best term, as turtles in this study they were more nomadic, and moved throughout a very large areas. Indeed, the home range size you mention of up to 181,342 ha, is not reflective of what the term ‘home range’ is typically intended to describe. This is so because for the study turtles, when they stop to feed, the food patch does not stay in the same place all the time, thus turtles are not really inhabiting a fixed home range, but rather an ephemeral, spatially dynamic, food rich area. I encourage the authors to think about this and propose a different, perhaps novel term for these ‘high seas foraging ranges’. Adding such a term would be a valuable contribution to the science of sea turtle movement analysis.

Third, relating to the above, for satellite tracked turtles, the home range (or in your case the high seas foraging range) of a turtle is only a subset of the entire track duration, except for when turtles are equipped with satellite transmitters after being captured within their foraging habitat. Otherwise, such as the case with turtles moving in the open ocean or in offshore, non-coastal waters, any given turtle track will usually have a portion of the track that is 'directional movement' as the turtle moved from point A to point B, and another portion of the track where they encounter food and slow their movements to forage in those localities. Often researchers use turning angles or travel speed filters to depict when a turtle has slowed down and reached its foraging area; the track portion that occurs outside of these foraging areas is not part of its home range or foraging range, but rather a transitional area, or movement area/corridor. In this paper, it is unclear in the methods which portion of the tracks were included in the home range analysis. Was it the entire track? Was it a subset of the track once the turtle slowed down? If the former, then inclusion of the entire track is not an accurate depiction of a home range or foraging range MCP. If you are going to include the entire track in the home range analysis then this approach needs to be thoroughly justified in the text. Instead it appears to me that the 50% PVC is used to depict the core area of foraging. If using the PVC 50% area to determine ‘foraging range’, this needs to be justified, considering that the vast majority of spatial data analyses use some sort of data filter to determine the area on a track that includes foraging activity. With that said, if you can show that the daily movement speeds of turtles occupying the 50% PVC is significantly slower than turtles outside these areas, this would be compelling support of the use of 50% PVC areas for depicting areas of foraging for a turtle in offshore waters. That being said, I am highly skeptical of this approach because the 50% PVC--as in the case of KOR0149--can be high density points not because a turtle had stopped there and spent much time, but rather because it moved back and forth through an area, never slowing down, yet because the turtle crisscrossed the area, the satellite location density is high, and the area is calculated mathematically as a core area, when in fact it is simply the 'crossroads' where a turtle keeps passing through. For me, a true core area would be more like what is depicted for KOR0151. This is a very important consideration and this caveat needs to be presented in the text if keeping the 50%PVC approach.

Fourth, and relating to PVC evaluations, the authors note that “We defined the 50 PVC as the area where turtles stayed for more than 15 days while foraging.”. Why was 15 days picked as a threshold? This seems arbitrary. It is quite possible that, due to the ephemeral nature of high seas food patches, that turtles could have encountered such areas an foraged for less than 15 days before moving to the next patch. Have the authors considered this? A much more robust technique would be to implement some sort of movement speed or turning angle filter to depict where on the movement path did turtles slow down, turn more, and likely forage.

Fifth, for the analysis of oceanographic variables, based on the paragraph starting on line 124, it is unclear if oceanography was measure for the entire track, or just the 50%PVC locations as is suggested in the following paragraph. If the latter, then this would be a great opportunity to further support the use of the 50% PVC area to depict foraging by comparing oceanographic variables in the 50% area vs. the remainder of the tracks.

These are my main concerns with the paper that need to be addressed before this manuscript is acceptable for publication. I hope the authors find these comments constructive, and I do believe that the information in this study is highly publishable, as soon as the authors correctly frame what the spatial parameters are that they are analyzing.

In addition, here area few inline text edits that may be helpful.

46 – change ‘since’ to ‘because’

53 – insert ‘studying’ after ‘for’

61-63 – it is unnecessary to list all the species of sea turtles in Japan, as this not relevant to this paper. Instead start with a statement that is true for both greens and loggerheads foraging and nesting in the region.

67 – change ‘on the other hand’ to ‘nevertheless’, which is a more appropriate term here and not common jargon.

70 – this line should read: “increased observation of loggerheads and green turtles at these distribution margins, studies on their habitat use are lacking.”

77 – add ‘subadult and adult’ in front of turtles names. This will build off their mention in the previous paragraph.

79 – this line should read: “provide essential information about their spatial biology, which can be useful…”

88 – this line should read: “identified as adults, and three turtles as potential…”

92 – give range in SST after mentioning temperature.

93 – for methods of attachment please describe the carapace cleaning and sanding preparation process

98 – last day ‘of’

103 – you should also mention which LCs were included specifically after “the high-accuracy data”

156 – should be “with movements ranging from [lat long coordinates]

J. Seminoff, 10 September 2023

Fig 1 legend, add “ ,respectively “ after “end points”

Fig 1 – very hard to see the release circles. Can these be made bigger? Also, each map should be labeled loggerhead and green turtle at top of respective maps to make it easier to interpret the figure. Also OK to have in caption.

6. PLOS authors have the option to publish the peer review history of their article (what does this mean?). If published, this will include your full peer review and any attached files.

Reviewer #1: No

Reviewer #2: **Yes: **Jeffrey Aleksandr Seminoff

---

## [Author Response · Author response to Decision Letter 0]

18 Jan 2024

Thank you for your loyal review. We've implemented most of the suggestions made by both reviewers, and are responding to some of their comments. We look forward to your review again.

---

## [Editor Report · Decision Letter 1]

24 Jan 2024

PONE-D-23-24778R1Habitat use of loggerhead (Caretta caretta) and green (Chelonia mydas) turtles in the northern limit of their distribution range of the Northwest Pacific OceanPLOS ONE

Dear Dr. An,

Thank you for submitting your manuscript to PLOS ONE. After careful consideration, we feel that it has merit but does not fully meet PLOS ONE’s publication criteria as it currently stands. Therefore, we invite you to submit a revised version of the manuscript that addresses the points raised during the review process.

We look forward to receiving your revised manuscript.

Kind regards,

Lee W Cooper, Ph.D.

Section Editor

PLOS ONE

I summarize the Journal Requirements below, which you have already considered in your most recent revision.

Journal Requirements: 

**Additional Editor Comments:**

Thank you for submitting your manuscript to PLOS ONE and thank you for addressing the reviewer comments, which I think you have done adequately. I am returning the manuscript to you, and ask that you address some comments and editing suggestions that I made using the comment function in the Adobe Acrobat software. For the most part, I have made suggestions that would improve the English language use in the text. As I mentioned, I think you have succeeded in addressing the reviewer comments regarding the scientific insights provided by your paper. Therefore, we invite you to submit a revised version of the manuscript that addresses the points raised by my review. I would also encourage you to consider acknowledging the two reviewers in your acknowledgements section, as I think their efforts helped improve the communication of your scientific findings.

---

## [Author Response · Author response to Decision Letter 1]

28 Jan 2024

We greatly appreciate your efforts in improving the quality of our manuscript. According to the suggestions, we revised it and all our edits can be viewed in the “track change” in the newly attached manuscript.

---

## [Editor Report · Decision Letter 2]

31 Jan 2024

Habitat use of loggerhead (Caretta caretta) and green (Chelonia mydas) turtles at the northern limit of their distribution range of the Northwest Pacific Ocean

PONE-D-23-24778R2

Dear Mr. An,

Thank you for making those final changes in the manuscript that I suggested. As a result, I'm pleased to inform you that your manuscript has been judged scientifically suitable for publication and will be formally accepted for publication once it meets all outstanding technical requirements.

Kind regards,

Lee W Cooper, Ph.D.

Section Editor

PLOS ONE

---

## [Editor Report · Acceptance letter]

26 Mar 2024

PONE-D-23-24778R2 

PLOS ONE

Dear Dr. An, 

I'm pleased to inform you that your manuscript has been deemed suitable for publication in PLOS ONE. Congratulations! Your manuscript is now being handed over to our production team.

Kind regards, 

on behalf of

Dr. Lee W Cooper 

Section Editor

PLOS ONE